# Visible light-induced switching of soft matter materials properties based on thioindigo photoswitches

Sarah L. Walden[1,2], Phuong H. D. Nguyen[3], Hao-Kai Li [4], Xiaogang Liu [4], Minh T. N. Le[3], Loh Xian Jun [5,6] ✉, Christopher Barner-Kowollik [1,2,7] ✉ & Vinh X. Truong [2,5] ✉

Thioindigos are visible light responsive photoswitches with excellent spatial control over the conformational change between their *trans-* and *cis-* isomers. However, they possess limited solubility in all conventional organic solvents and polymers, hindering their application in soft matter materials. Herein, we introduce a strategy for the covalent insertion of thioindigo units into polymer main chains, enabling thioindigos to function within crosslinked polymeric hydrogels. We overcome their solubility issue by developing a thioindigo bismethacrylate linker able to undergo radical initiated thiol-ene reaction for step-growth polymerization, generating indigo-containing polymers. The optimal wavelength for the reversible *trans-/cis-* isomerisation of thioindigo was elucidated by constructing a detailed photochemical action plot of their switching efficiencies at a wide range of monochromatic wavelengths. Critically, indigo-containing polymers display significant photoswitching of the materials' optical and physical properties in organic solvents and water. Furthermore, the photoswitching of thioindigo within crosslinked structures enables visible light induced modulation of the hydrogel stiffness. Both the thioindigo-containing hydrogels and photoswitching processes are non-toxic to cells, thus offering opportunities for advanced applications in soft matter materials and biology-related research.

Photoswiches are light absorbing molecules that can be reversibly interconverted to different isomers with distinct chemical and physical properties[1,2]. The photo-induced reversible changes in their properties – coupled with the exquisite spatial and temporal light based control – make them an excellent stimuli-responsive platform for a wide range of applications in actuators[3], optics[4–7], molecular motors[8,9], and photobiology[8,10]. Within the current toolbox of photoswitches, the most widely used chromophores are derivatives of azobenzenes[11,12], diarylethenes[6,7,13,14] spiropyrans[5,15,16], and bisimines[17]. In the biosciences, azobenzene derivatives are often the preferred choice, mainly due to

[1]Centre for Materials Science, Queensland University of Technology (QUT), 2 George Street, Brisbane QLD 4000, Australia. [2]School of Chemistry and Physics, Queensland University of Technology (QUT), 2 George Street, Brisbane QLD 4000, Australia. [3]Department of Pharmacology and Institute for Digital Medicine, Yong Loo Lin School of Medicine, National University of Singapore, Singapore 117600, Republic of Singapore. [4]Fluorescence Research Group, Singapore University of Technology and Design, 8 Somapah Road, 487372 Singapore, Republic of Singapore. [5]Institute of Materials Research and Engineering (IMRE), Agency for Science, Technology and Research (A*STAR), 2 Fusionopolis Way, Singapore 138634, Republic of Singapore. [6]Institute of Sustainability for Chemicals, Energy and Environment (ISCE2), Agency for Science, Technology and Research (A*STAR), 1 Pesek Road, Jurong Island, Singapore 627833, Republic of Singapore. [7]Institute of Nanotechnology (INT), Karlsruhe Institute of Technology (KIT), Hermann-von-Helmholtz-Platz 1, 76344 Eggenstein-Leopoldshafen, Germany. ✉e-mail: lohxj@imre.a-star.edu.sg; christopher.barnerkowollik@qut.edu.au; vinh_truong@isce2.a-star.edu.sg

their geometric change induced by *trans-/cis-* isomerization[1,10,11]. Such molecular motion enables substantial conformational changes in their surrounding environment, allowing biologists to discretely photo-regulate mechanical functions in biological settings for investigation of e.g., cell materials interaction (mechanotransduction)[18,19], or the transport of bioactive components across cellular environments[20–22]. However, the spatial control exerted by azobenzenes is limited by a wide distribution of geometries of the *cis-* isomers, due to the low barriers of rotation around the single C-N bond[23]. This limitation is rooted in the main pathway of azobenzene *trans-/cis-* isomerization, where the conformational change occurs via rotation of the C-N=N-C dihedral torsion angle instead of the inversion of the C-N=N in-plane bending angle[24]. Such a limitation is also encountered in other photoswitches derived from fulgimides or stilbenes[25,26].

In contrast, (thio)indigo derivatives offer exceptional spatial control over conformational change, enabled by a 180° rotation around the central C=C bond when the molecules interconvert between their *trans-* and *cis-* isomers (Fig. 1)[27–29]. The absorption of the *cis-*form is blue-shifted compared to the *trans-*form, due to the non-bonding interaction between the sulfur/nitrogen and oxygen atoms[30] rather than the sterically induced distortion from planarity and impaired conjugation encountered in azobenzene- and stilbene-based photoswitches[24,26]. These features make (thio)indigos excellent candidates for molecular machinery, providing robust spatial control over an attached molecular payload[10,31–33]. However, despite the discovery of their photochromism over 100 years ago[34,35] and the extensive exploration of photoisomerization mechanisms[27,30,36–38], applications of (thio)indigos in soft matter materials remain elusive. The very few reports on thioindigos' utilisation mostly focused on dye-doping of ferroelectric liquid crystal displays[31,32], and solid supports[39–41], or a molecular tweezer operating at micromolar concentration for the capture and release of metal ions[42]. Similarly, applications of indigo in polymeric materials were only recently demonstrated as light responsive dopants in polymer thin film[43].

A major limiting factor of (thio)indigos is their low solubility in almost all conventional organic solvents and polymers[44], preventing the covalent incorporation or compatibilization of such entities into polymeric materials. Indeed, the rigid and almost planar structure of both isomers drives the molecules towards aggregation, significantly reducing their mobility and reactivity[27]. Thus, suitable side chains such as alkyl and oligomeric siloxane have been introduced – via demanding synthetic procedures – to decrease the aggregation, yet the use of such chromophores is still limited to acting as dopants in liquid crystal host[31,32,45] or polymer film for fluorescent optical recording[46]. Recent attempts to modify thioindigo scaffolds have resulted in water soluble

adducts by virtue of sulfonic acid residues directly attached to the benzyl rings. However, their water solubility is still within micromolar concentration (1-5 μM)[33,47]. Any higher concentration was reported to result in major solute-solute interaction, and associated aggregation that causes broadening of the UV-Vis absorption spectra, as well as the failure to identify specific isomers in nuclear magnetic resonance (NMR) analysis[33]. Therefore, suitable solubilizing strategies are critical to establish the utility of thioindigos in aqueous environments, paving the way for their applications in chemical and mechanical biology, as well as smart soft matter materials.

Herein, we pioneer a strategy towards the integration of the thioindigo function into polymer main chains, enabling the assessment of the photoswitching of the thioindigo-containing macromolecules in various solvents, including water (Fig. 1). Our approach employs a polymerizable thioindigo linker that can be readily synthesised on the gram scale. The photoswitching of the small molecule is thoroughly investigated, using a tuneable laser system to reveal the optimal wavelength irradiation for photoisomerization via an action plot analysis. Critically, we develop a strategy to incorporate thioindigo into biocompatible crosslinked hydrogel structures, demonstrating that thioindigo photoisomerization can result in the modulation of hydrogels' stiffness by light in the visible region.

## Results

### Assessment of photoswitching of a thioindigo bismethacrylate crosslinker

Our approach for the thioindigo linker was designed such that the reagent contains methacrylate function that can participate in conventional free-radical polymerization. Thus, we developed a facile protocol for the gram-scale preparation of a bismethacrylate thioindigo (*trans*-**1**, Fig. 2a) with an overall yield of 15.4%, and with a simple purification procedure that does not require column chromatography (refer to SI, Section 2.1 for the synthesis procedure and characterisation of *trans*-**1**). Despite having the methacrylate and ester substitution on the aromatic rings, thioindigo **1** is insoluble in all polar organic solvents including methanol, ethanol, dimethyl sulfoxide (DMSO), and dimethylformamide (DMF). When screening conventional organic solvents, we found that chloroform is the only solvent that dissolves **1** at concentrations up to 5 mM (3 mg·mL$^{-1}$), which allows for UV-Vis absorption and routine NMR analysis. The UV-Vis absorption spectrum of **1** displays a peak at 540 nm, similar to that of the previously reported thioindigos (Fig. 2b). Irradiation of solution **1** at $\lambda_{max}$ = 540 nm results in the decrease of the 540 nm absorbance and an increasing appearance of the blue-shifted peak at 490 nm, with an isosbestic point at 503 nm, indicating the conversion to the *cis*-isomer. Irradiation of the *cis*-isomer solution by $\lambda_{max}$ = 490 nm resulted in the reversion of the absorbance to the *trans*- species. The UV-vis absorbances of *trans*- and *cis*-thioindigos are better resolved than isomers of azobenzene[48], for the reasons explained above[27,37]. The reversible *trans-/cis*-photoswitching can be repeated for more than 10 cycles without any significant changes in the absorbance spectra of both isomers (Fig. 2c), indicating that no major photodegradation of thioindigo **1** in CHCl$_3$ occurred during alternating irradiation of green (540 nm) and blue (490 nm) wavelengths.

To determine the optimum wavelength for the photoisomerization that affords the strongest enrichment of each isomer, we employed a nanosecond pulsed, tuneable optical parametric oscillator (OPO) laser coupled with a UV-Vis spectrometer to screen the activation wavelength. By tracking the in-situ absorbance measurements during irradiation, we were able to determine the quantum yield of isomerisation and *cis-/trans-*ratio at discrete wavelengths across a broad spectrum that covers the absorption spectra of both isomers in a photochemical action plot[49]. Initially, we compare the wavelength dependent photoisomerisation quantum yield to the absorption spectrum. Individual experimental data of the photoisomerisation at

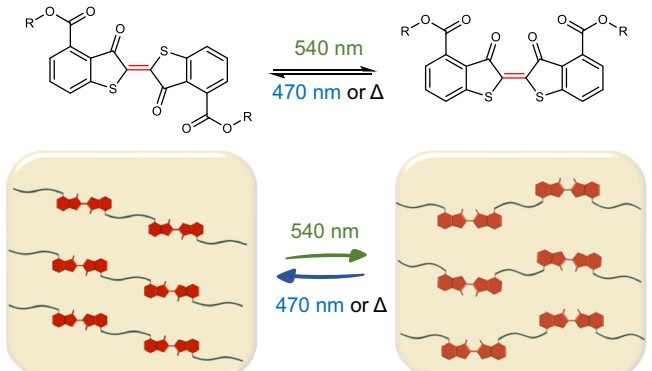

**Fig. 1 | Schematic overview of photoswitching.** Thioindigo moieties undergo reversible photoisomerization enabled by green and blue light, leading to a spatial conformational change of the attached molecular payload by a 180° rotation around the C = C bond. Incorporation of the thioindigo into the polymer main chain affords photoswitching of the soft matter materials properties.

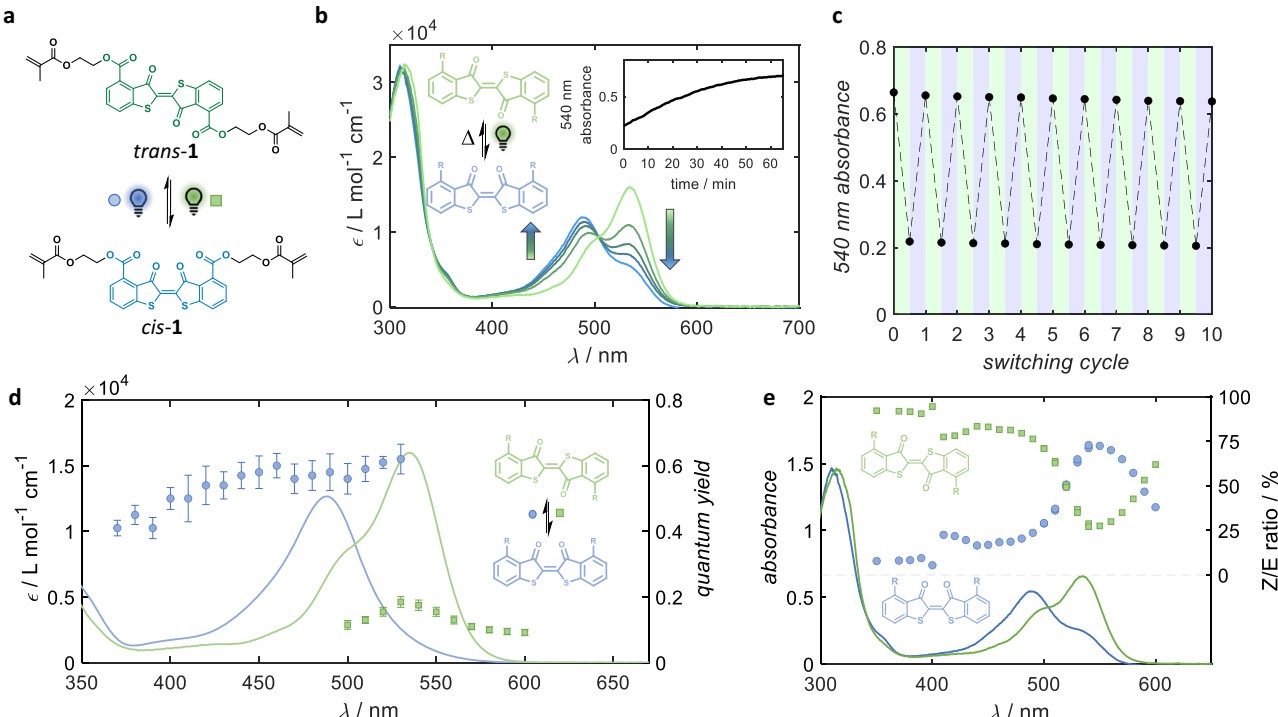

**Fig. 2 | Photoisomerization of a polymerizable thioindigo linker. a** Reaction scheme of the photoswitching of thioindigo bismethacrylate (in *trans*-form, green line) by green ($\lambda_{max}$ = 540 nm) and blue ($\lambda_{max}$ = 490 nm) light; **b** UV-vis spectra of **1** under green light irradiation showing the *trans*- to *cis*-conversion, which can be reverted by blue light; the inset displays *cis*-/*trans*- conversion kinetics at ambient (24 °C) temperature in the dark; **c** cyclic photoswitching of **1** for up to 10 cycles by alternative green and blue light irradiation; **d** wavelength-dependent reversible *trans*-**1**/*cis*-**1** switching rate, normalised by incident photon flux, and overlayed with the absorbance spectra of *trans*-**1** (green line) and *cis*-**1** (blue line); the inset displays a zoom-in wavelength window of 550-620 nm, the total error is the sum of the relative errors arising from the numerical fits and experimental fluctuations in laser power; **e** wavelength-dependent rates of the reversible photoswitching overlayed with the spectra of the *trans*- and *cis*-isomers.

each wavelength and procedure for determining the quantum yield is provided in Section 1.5 of the Supplementary Information. The wavelength dependent quantum yields of *trans*- to *cis*- and *cis*- to *trans*-isomerisation are depicted in Fig. 2d as green circles and blue squares, respectively. The largest recorded quantum yields were 0.18 ± 0.02 and 0.62 ± 0.05 for *trans*- to *cis*- isomerisation and *cis*- to trans- isomerisation, respectively, both recorded at 530 nm. As one would intuitively expect, the quantum yield does not vary significantly with the employed wavelength. However, the larger quantum yield of the *cis*- to *trans*-isomerisation suggests that attempts to achieve *trans*- to *cis*-isomerisation with shorter wavelengths, where the absorptivity of the *cis*-isomer becomes significant (<530 nm), inevitably also triggers efficient back-isomerisation. Fortunately, photoinduced switching was found to persist up to $\lambda_{max}$ = 600 nm, despite thioindigo **1** showing minimal absorbance. These results alone, however, are not sufficient to inform the best wavelengths for photoisomerization. It is important to also consider the *cis*-/*trans*-ratio presented in Fig. 2e. For *trans*- to *cis*-isomerisation, the highest *cis*-/*trans*- ratio of 72.5% (blue circles, Fig. 2e) was achieved at wavelengths around 550 nm (15 nm red-shifted from the absorption maximum). Notably, almost no *trans*- to *cis*-isomerization occurred in the mild ultraviolet regime (350-400 nm). For *cis*- to *trans*-isomerisation, high *trans*-/*cis*- ratios (green squares, Fig. 2e) exceeding 80% are observed in the wavelength region between 440-470 nm, which is blue-shifted relative to the absorption maximum. The maximum recorded *trans*-/*cis*- ratio was 94.5% recorded at 400 nm. Taking into account both quantum yields and isomer ratios, it was determined that the ideal wavelengths for *trans*- to *cis*-isomerisation and *cis*- to *trans*-isomerisation are 540-550 nm and 450-470 nm, respectively.

The *cis*-form of thioindigo **1** was observed to spontaneously revert to *trans*-isomer at ambient temperature (*ca.* 24 °C), with a thermal half-life of 24 minutes (Fig. 2b), similar to those of other previously reported thioindigos in aprotic solvents[36,47]. The relatively fast thermal isomerisation prevented us from obtaining a clear spectrum of the *cis*-isomer, since a 10-20 min interval is required before an NMR spectrum can be recorded post-irradiation. In addition, the concentration required for routine NMR measurements is much higher than that of UV-Vis absorbance measurements (2 mg·mL⁻¹ versus 0.2 mg·mL⁻¹), leading to a higher thermal *cis*- to *trans*- isomerization rate[33]. We also observed precipitation during green light irradiation of *trans*-**1** due to the strong aggregation effects of the *cis*- and *trans*- isomers in chloroform. Consequently, we were only able to obtain an NMR spectrum of a *cis*-**1**/*trans*-**1** ratio of 3/7, by integration of the corresponding aromatic resonances (Supplementary Figure 11).

## Photoswitching of thioindigo-containing polymer in conventional organic solvents and water

Following the determination of the optimal wavelength for photoisomerization of crosslinker **1**, we proceeded to integrate the crosslinker into a polymer structure. Surprisingly, the methacrylate moieties on thioindigo **1** are highly resilient against free radical activation, as no change in ¹H NMR spectrum was observed even when a solution of **1** and azobisisobutyronitrile (AIBN, 1 mol%) in chloroform was heated under reflux at 70 °C for 48 h. Attempts to use the methacrylate group as an acceptor in Michael thiol additions were also unsuccessful, since conventional base catalysts, including pyridine and triethylamine, did not initiate the thiol-methacrylate reaction; stronger catalysts such as 1,8-diazabicyclo(5.4.0)undec-7-ene (DBU) and dimethylphenylphosphine led to the decomposition of thioindigo scaffold and discoloration. The breakthrough reported herein was ultimately achieved by free radical initiated thiol-ene chemistry of the methacrylates of **1** with thiols using AIBN and at 65 °C in CHCl₃.

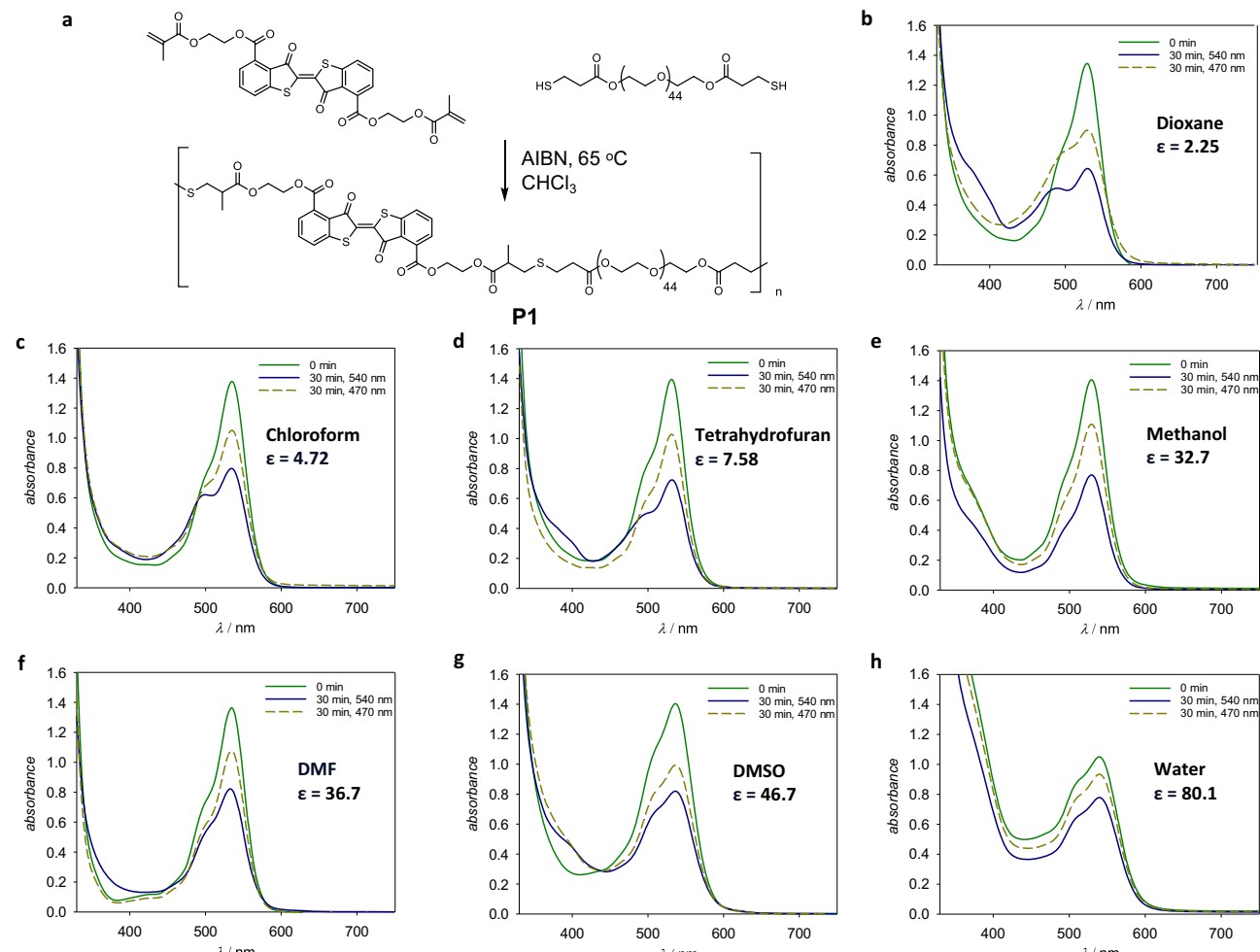

**Fig. 3 | Photoswitching of thioindigo within a PEG chain. a** Reaction scheme of the synthesis of thioindigo-containing PEG. **b–h** UV-vis spectra of polymer **P1** in a range of solvents with increasing dielectric constant ε before and after irradiation with green ($\lambda_{max}$ = 540 nm) and blue light ($\lambda_{max}$ = 470 nm).

For polymerization, we selected poly(ethylene glycol) (PEG) as the water-soluble precursor and modified the end-group to confer the thiol function, **PEG$_{44}$-(SH)$_2$** ($M_n$ = 2176 g·mol$^{-1}$, $Đ$ = 1.03, refer to the SI, Section 2.2 for the synthesis procedure). Thioindigo-containing PEG was subsequently prepared by free radical thiol-ene step-growth polymerization (Fig. 3a), forming polymer **P1** with a molar mass of 13,058 g·mol$^{-1}$ ($Đ$ = 2.73). Importantly, **P1** is soluble in a wide range of polar solvents, enabling the study of photoswitching by UV-Vis absorbance measurements. Based on our action plot study, we selected LED light sources with emissions centred at 540 and 470 nm for the reversible photoisomerization of **P1** (Supplementary Figure 6). Compared to the chloroform solution of small molecule thioindigo bismethacrylate *trans*-**1**, **P1** displayed a similar absorption spectrum in all investigated solvents, and green light ($\lambda_{max}$ = 540 nm) irradiation on **P1** solutions led to the decrease of the 540 nm peak (Fig. 3b–e). However, the accompanying absorbances of the *cis*-isomer are not well resolved, suggesting lower *trans*- to *cis*-conversion ratios compared to the photoisomerization of small molecule *trans*-**1**.

Inspection of Fig. 3 reveals that the absorbance at 490 nm decreases with increasing solvent dielectric constant after green light irradiation. Specifically, in non-polar aprotic solvents such as dioxane (ε = 2.25) and chloroform (ε = 7.58), the *cis*-/*trans*- absorbance ratios at the photostationary state decrease from 0.57 to 0.43, respectively. The *cis*- to *trans*-conversion can also be induced by blue light at $\lambda_{max}$ = 470 nm, reaching a photostationary with higher *trans*-isomer content compared to green light treatment. The absorbance spectra of

**P1** in polar solvents (those with ε values > 30) indicate complex solute–solute interactions and aggregations – especially in polar protic solvents such as methanol and water, as can be seen from the broadening of the absorbance (Fig. 3e and h). These interactions inevitably limit the assessment of the *cis*-/*trans*- ratios at the photostationary states in highly polar solvents. However, the decrease and increase of the 540 nm peak, characteristic of the thioindigo photoisomerization, can be induced by green and blue light irradiation, respectively, suggesting photoswitching of the solutes. The photoisomerization in these protic solvents can be induced for up to 10 cycles of green/blue light irradiation, indicating no major photodegradation occurring (Supplementary Figure 14). Thus, our investigation offers a critical answer for the utility of thioindigo in soft matter materials: thioindigo moieties – when covalently incorporated within the polymer main chain – display appreciable photoisomerization in both aprotic and protic polar solvents.

Interestingly, solutions of **P1** in water at c ≥ 5 wt% display thermoresponsive gelation behaviour that can be photo-modulated. Specifically, an aqueous solution of **P1** at c = 5 wt% formed a solid gel at 37 °C, reaching a storage modulus of *ca*. 1000 Pa. Rheological assessment of the hydrogel showed a decrease in the storage modulus to *ca*. 100 Pa under green light ($\lambda_{max}$ = 540 nm) irradiation (Fig. 4a). Blue light ($\lambda_{max}$ = 470 nm) irradiation of the mixture resulted in the recovery of the *G'* value to approximately 60% of the initial value before green light irradiation. The photoswitching of the *G'* value can be induced for several cycles, corresponding to the reversible *trans*-/*cis*-

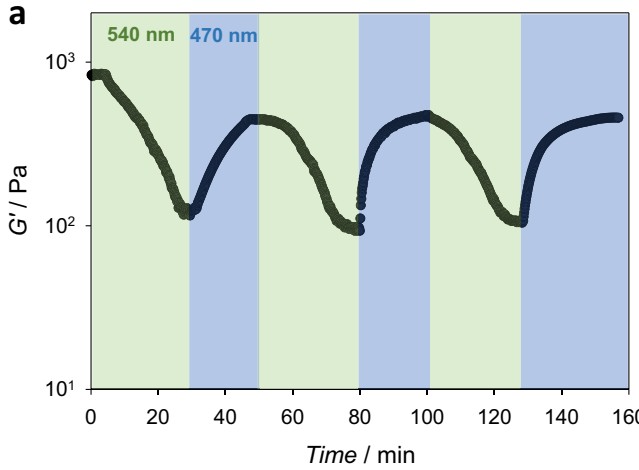

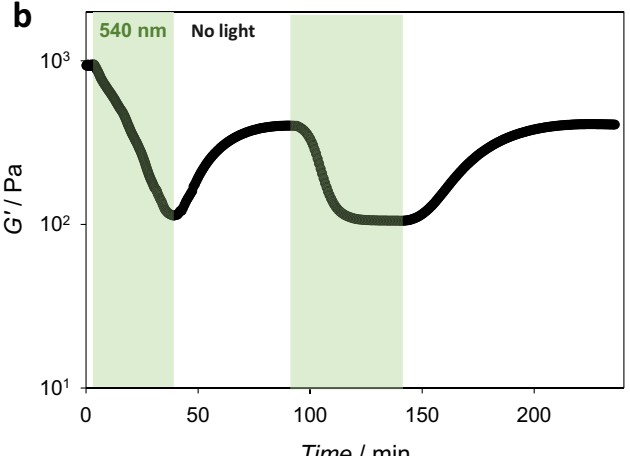

**Fig. 4 | Photoswitching of a thermoresponsive hydrogel. a** Rheological data demonstrating the change in storage modulus $G'$ of a **P1** hydrogel (5 wt% in deionized water) as a function of time at 37 °C under alternating green and blue light irradiation; **b** Rheological data of P1 hydrogel under green light irradiation followed by spontaneous thermal recovery.

isomerization of the thioindigo units. Furthermore, the $G'$ value can recuperate spontaneously at 37 °C, to a value similar to those recorded under blue light irradiation (Fig. 4b).

### Computational calculations of photoswitching of thioindigo

We subsequently employed computational calculations to rationalize the decrease in photoswitching efficiency of the thioindigo with increasing solvent polarity. Initially, we optimized the geometries of thioindigo in both its *trans-* and *cis-* forms and obtained the energy levels of the frontier molecular orbitals, which encompass the highest occupied molecular orbital (HOMO) and the lowest unoccupied molecular orbital (LUMO; Fig. 5a). The *trans-* isomer affords a lower electronic gap of 4.689 eV, in comparison to that of the *cis-* isomer (4.888 eV). The calculated relative Gibbs free energy of the *trans-* and *cis-* forms of thioindigo (Fig. 5b) indicates that the *trans-*isomer is more stable (by approximately 0.4 eV) and exhibits longer $\lambda_{abs}$: 472 nm and 444 nm for the *trans-* and *cis-* isomers, respectively. We also noted that the *trans-*isomer possesses a lower dipole moment (4.39 Debye) than that of the *cis-*isomer (5.87 Debye). It should be noted that the M06-2X functional typically overestimates the excitation energy, leading to a blue shift compared to experimental data. However, our calculations demonstrate that the *trans-*form possesses longer $\lambda_{abs}$, which are in good agreement with the experimental data. Subsequently, we computed the potential energy surface of thioindigo in its ground state,

varying with intramolecular rotation along the central bond (Fig. 5c). By incorporating constraints into our calculations, we expect that the actual energy barrier is even lower than the values presented (~2.5 eV). Nonetheless, the results unequivocally establish the enhanced stability of the *trans-*isomer. These findings provide a clear rationale for the intrinsic inclination of the *cis-*isomer to spontaneously transition to the *trans-*isomer in the absence of green light irradiation.

Upon photo-excitation, thioindigo can undergo intramolecular rotation along the central bond as a function of θ, facilitating both the *trans-* to *cis-* transformation (with an energy barrier of 0.423 eV) and the *cis-* to *trans-* transformation (with an energy barrier of 0.37 eV; Fig. 5c). It is important to note that the S1 state of the trans conformation is associated with a π-π* transition. After excited-state geometry optimization, the S1 state of the *cis-* conformation is characterized by an n-π* transition, which is highlighted in orange (Fig. 5d). Our calculations reveal that protic solvents, such as methanol, can hinder the photoswitching of thioindigo by forming hydrogen bonds. Specifically, the evaluated hydrogen bonding energy between thioindigo and methanol molecules in the excited state is approximately 0.4 eV per hydrogen bond (Fig. 5e). When two hydrogen bonds form around the thioindigo, the bonding energy increases to about 0.8 eV. These hydrogen bonds act as barriers to the rotation of thioindigo in the polymer chain upon photoexcitation – a key step in the photoswitching process. Given the limited free volume within the polymer chain conformation, thioindigo must first break these hydrogen bonds before undergoing the required intramolecular rotation. Consequently, the presence of hydrogen bonds in protic solvents, such as methanol and water, suppresses the photoisomerization. In cases where there is sufficient free volume to accommodate the intramolecular rotation of the thioindigo-methanol/water complex, the increased volume of the complex (compared to the thioindigo monomer) further amplifies the rotational resistance within the polymer chain. It is expected that high-polarity solvents tend to promote the prevalence of the *trans-*isomer, consequently contributing to a lower *cis-*isomer content in the photostationary state. Our computational results agree with earlier quantum chemical calculations of a water soluble thioindigo molecule[27,47], supporting the understanding that the *cis-*conformer of thioindigo exhibits considerably lower stability than the *trans-*conformer in polar solvents. Furthermore, the *cis-* form exhibits a red-shifted absorbance when in a polar solvent compared to a non-polar solvent, as previously reported[47].

### Incorporation of thioindigo into click-crosslinked PEG hydrogels for visible light induced photoswitching of physical properties

Having confirmed the photoswitching of the thioindigos within polymer chains in an aqueous environment, we subsequently designed a strategy to incorporate the photolabile function into hydrogels with robust mechanical strength, enabling photo-modulation of the hydrogels' stiffness. The very low solubility of the thioindigo bis-methacrylate *trans*-**1** prevents the direct mixing of the reagent in a resin formulation for crosslinking. Thus, we initially synthesised a polymer precursor **P2** from **1** and **PEG$_{12}$-(SH)$_2$** by free radical thiol-ene polymerization in CHCl$_3$ (Fig. 6a) The intermediate product **P2** has sufficient solubility in DMF at a concentration suitable for polymer crosslinker with a multi-arm linker (>1 g·mL$^{-1}$). **PEG$_{12}$-(SH)$_2$** was used in excess, ensuring the remaining thiol function can be employed in the subsequent crosslinking, which occurs via catalyst-free nucleophilic thiol-propiolate addition with a PEG$_{448}$-(propiolate)$_4$ in DMF (Fig. 6a)[50]. The resultant organogels were subsequently treated with excess water over 24 h to obtain fully swollen hydrogels. We were thus able to covalently embed various amounts of thioindigo within the network structure, forming a series of hydrogels with an increasing weight ratio of thioindigo over the PEG component (Table 1), from 0.55 to 5.5 wt%.

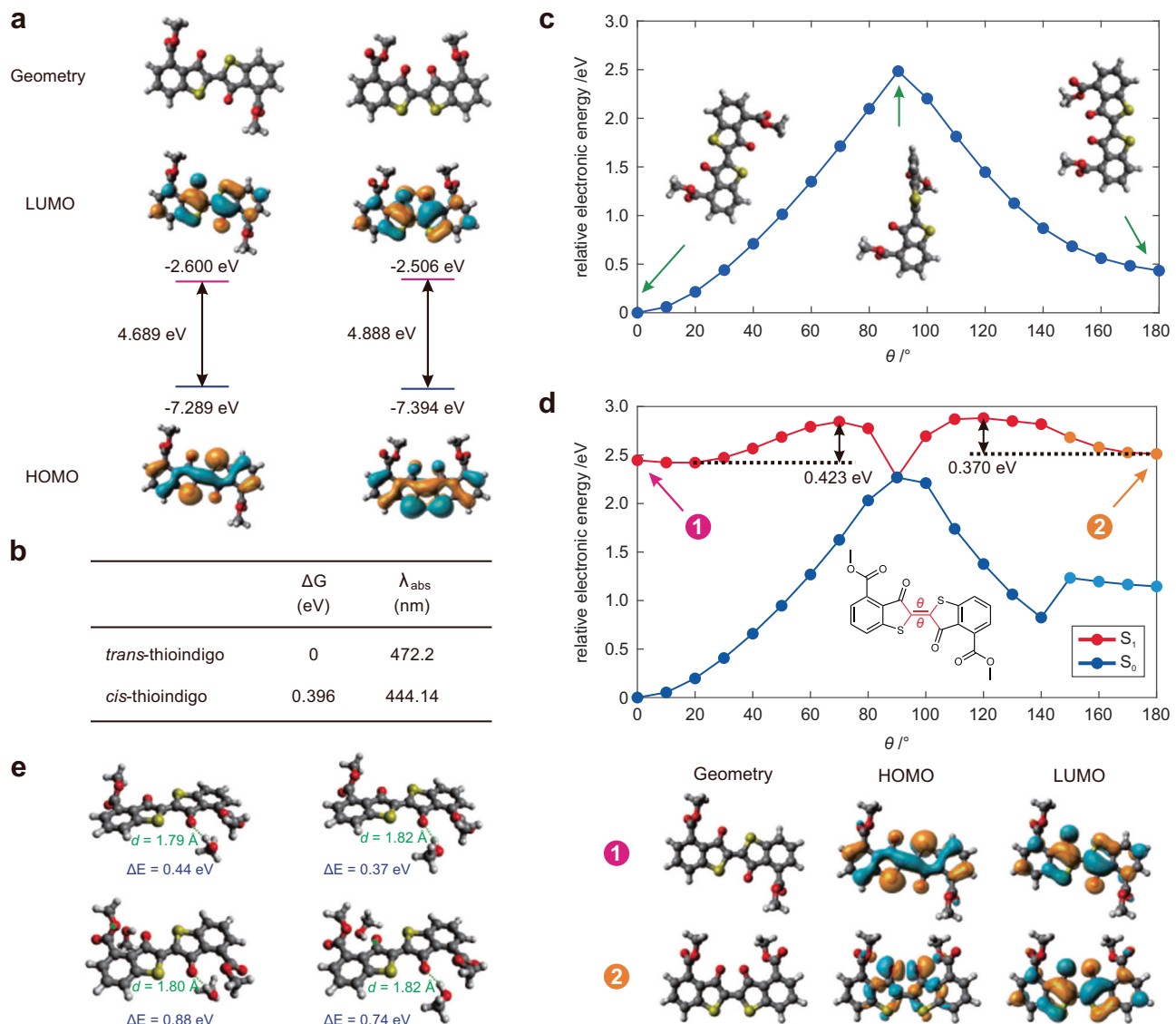

**Fig. 5 | Chemical quantum calculations of the photoisomerization of thioindigo. a** Optimized geometries and the corresponding frontier molecular orbitals of the trans- (the left panel) and cis- forms (the right panel) of photoswitch thioindigo in the ground state. The insets show the energy levels of frontier molecular orbitals and the corresponding electronic gaps (level of theory: M06-2X/def2-TZVP in dichloromethane; **b** Calculated relative Gibbs free energy (ΔG), peak UV-vis absorption wavelengths (λ$_{abs}$), and ground state dipole moment (μ) of the trans- and cis-forms of thioindigo; **c** Potential energy surface (PES) in the ground state (S$_0$), as a function of θ. **d** Potential energy surface (PES) in the excited state (S$_1$) and the ground state (S$_0$), as a function of θ, based on optimized geometries in the S$_1$ state (the top panel), and the optimized geometries and the corresponding frontier molecular orbitals of the trans- and cis-forms of thioindigo (the bottom panel). The inset highlights the dihedral angle θ, representative molecular structures on the PES, as well as the energy barriers. **e** Optimized geometries of thioindigo-methanol complex in the first excited state (S$_1$). The insets show the hydrogen bond distances and associated energy. Level of theory: M06-2X/def2SVP in dichloromethane for (**c**) to (**e**).

These materials (**Gel$_{T1}$** to **Gel$_{T5}$**) allow us to systematically investigate the effect of thioindigo and its photoisomerization on hydrogels' properties, compared to a PEG-based hydrogel (**Gel$_{T0}$**) with no thioindigo.

We observed a considerable change in the optical and physical properties even with the small amount of thioindigo incorporated into the network structures. Specifically, with 0.55 wt% incorporation of thioindigo, a clear red hydrogel was formed. Increasing the weight content of the thioindigo to ≥1.6 wt% resulted in the formation of opaque and deep red hydrogels, as compared to the clear and transparent hydrogel with no thioindigo **Gel$_{T0}$** (Fig. 6b). Notably, we observed a decrease in the swelling ratio $Q_{equilibrium}$ (Table 1) and the associated increase in hydrogels' stiffness (Supplementary Figure 11) at the fully swollen state with increasing thioindigo content, despite the decrease in the thiol molar stoichiometry for thiol-propiolate

crosslinking (lower crosslinking density in hydrogels with higher thioindigo content). These results indicate the significant physical aggregation of the thioindigo moieties, enhancing the mechanical properties of the networks. Interestingly, UV-Vis spectra of **Gel$_{T5}$** after green light treatment showed a marked decrease in the 540 nm peak, similar to the change in absorbance of the thioindigo in polymer thin film[46]. Under blue light irradiation, the 540 nm absorbance was observed to recover, indicating the reversed photoisomerization.

The hydrogel with the highest thioindigo content (5.5 wt%, **Gel$_{T5}$**) showed variations in stiffness in response to both temperature change and light irradiation (Fig. 7a). In particular, **Gel$_{T5}$** displayed an increase in G′ value from 4.1 kPa to 4.7 kPa when the temperature was increased from 24 °C to 37 °C. When the hydrogel was subjected to green light irradiation at 37 °C, the G′ value decreased to ca. 3 kPa, and reverted to 3.8 kPa under blue light irradiation (Fig. 7b). The photo-modulation of

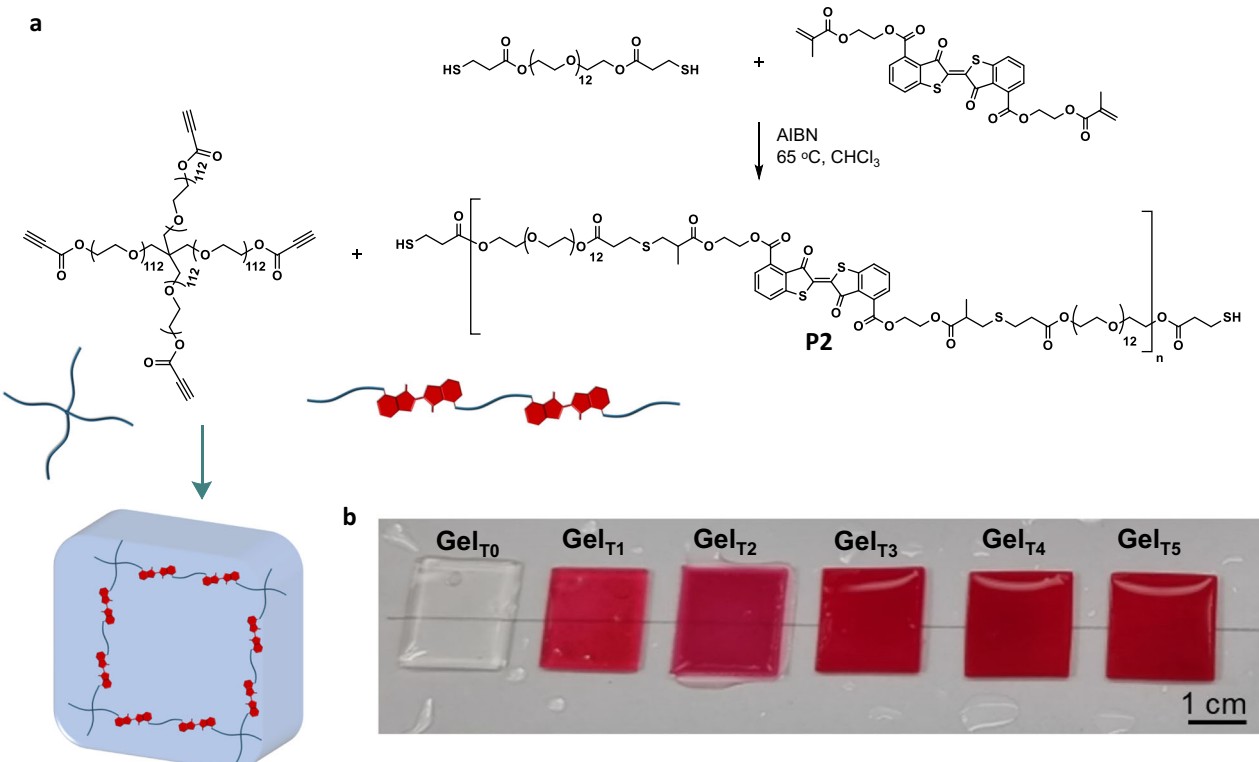

**Fig. 6 | Thioindigo-containing hydrogels. a** Reaction scheme of precursor synthesis and representative polymer crosslinking process via thiol-propiolate nucleophilic addition to form thioindigo-containing PEG-based hydrogels; and **b** photographs of the hydrogels with increasing amount of incorporated thioindigo in their fully swollen stages.

**Table 1 | Summary of precursor and PEG crosslinker concentrations used in the preparation of hydrogels with a range of thioindigo contents**

| | trans-1 (M) | $[PEG_{600}(SH)_2]$ (M) | $[PEG_{20k}(propiolate)_4]$ (M) | Trans-1 wt% | $Q_{equilibrium}$ [a] |
|---|---|---|---|---|---|
| $Gel_{T0}$ | 0 | 20 | 10 | 0% | 91.3% ± 0.4 |
| $Gel_{T1}$ | 2 | 20 | 10 | 0.55% | 89.5% ± 0.5 |
| $Gel_{T2}$ | 4 | 20 | 10 | 1.01% | 88.6% ± 0.7 |
| $Gel_{T3}$ | 6 | 20 | 10 | 1.66% | 85.1% ± 1.1 |
| $Gel_{T4}$ | 8 | 20 | 10 | 2.2% | 83.9 ± 1.2 |
| $Gel_{T5}$ | 10 | 20 | 10 | 5.5% | 81.4% ± 1.5 |

[a]Equilibrium swelling ratio, calculated by $(w_{swollen\,gel}\text{-}w_{dry\,gel})\cdot100\%/w_{swollen\,gel}$.

the hydrogel's stiffness by alternating green and blue light irradiation can be repeated for up to 10 cycles under rheological assessment, similar to the photo-responsiveness displayed by physically cross-linked hydrogel prepared from **P1** (Supplementary Figure 11). Likewise, the green light-induced softened hydrogel slowly recovered its storage modulus in the dark. We note that the kinetics of photo-induced switching of storage modulus is slower azobenzene-containing PEG hydrogels[51], which is attributed to the hydrogen bonding restriction of *trans-* to *cis-*rotation in polymer network as shown in chemical quantum calculations. In addition, similar to the photoswitching behaviour of azobenzene hydrogel, the G' value of **GelT₅** did not return to the original value before light irradiation[51], which is attributed to the aggregation and stabilization of the *cis-* conformer. The other hydrogels **Gel_{T3}** and **Gel_{T4}** also displayed switching of the stiffness, by 3% and 8% variation in *G*' value, respectively in response to the changes in green and blue light irradiation (Supplementary Figure 11). Hydrogels **Gel_{T0}**, **Gel_{T1}** and **Gel_{T2}** did not show any changes in the storage modulus when subjected to light irradiation, and only a minor increase in stiffness at elevated temperatures.

In addition to the decrease in the modulus values, **Gel_{T5}** also exhibited an increase in swelling ratio under green light irradiation, from 81.4% to 85.2%, accompanied by an increase in hydrogel size in water (Fig. 7a). The softening and increased swelling effect are attributed to the *cis-* conformer having a higher dipole moment (μ = 5.87 Debye) compared to the trans- conformer (μ = 4.39 Debye), and thus the *cis*-enriched hydrogel network is more polar compared to the *trans*-enriched network, enabling additional water update. We note that in non-polar solvents and when there is no significant change in the polarity of the photoswitched conformers, the unidirectional out-of-equilibrium rotation of the photoswitch could result in a network contraction[52,53] instead of expansion.

**Biocompatibility of hydrogels**

Biocompatibility is an important parameter for materials intended for cell-related research and biomedical applications. In order to assess whether the hydrogel materials and photoswitching in the current study are biocompatible, Annexin V and Propidium Iodide (PI) staining were performed to evaluate apoptosis of cells exposed to hydrogels

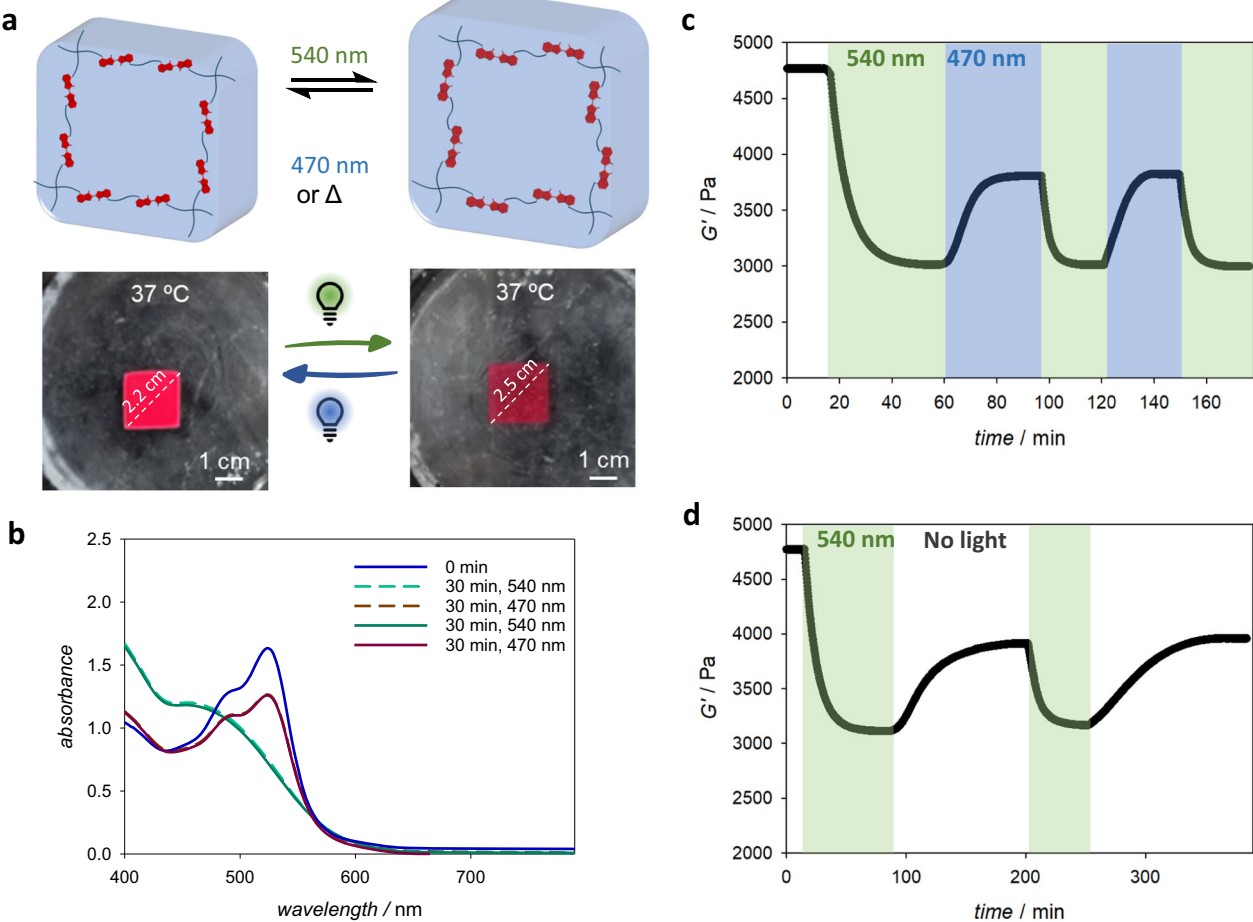

**Fig. 7 | Visible light-induced switching of hydrogels stiffness. a** Representative scheme of photoswitching of the inserted thioindigo isomers with different dipole moments within hydrogel structure (top), and images of **Gel$_{T5}$** in water at 37 °C following green ($\lambda_{max}$ = 540 nm) or blue ($\lambda_{max}$ = 470 nm) LED light irradiation for 30 min, displaying expansion and contraction; **b** UV-vis spectra of **Gel$_{T5}$** at 37 °C following green and blue light irradiation over 30 min of each treatment; **c** Rheological data of the photoswitching of **Gel$_{T5}$** under alternating green and blue light irradiations and **d** green light irradiation with intermittent thermal recovery.

and *trans-/cis-* photoswitching process. **Gel$_{T5}$**, which contains the highest amount of thioindigo and displayed a significant change in stiffness under green light irradiation, was selected for biocompatibility assessments. Figure 8a indicates that HEK-293T cells incubated with hydrogels before and after irradiation with a LED green light ($\lambda_{max}$ = 540 nm) maintained their polygonal morphology similar to the untreated control. In contrast, HEK-293T cells incubated with the positive control (DMSO) displayed changes in cytoplasmic morphology. Particularly, they appeared as clusters of cells with a rounded shape, suggesting the cytotoxic effects of DMSO on the cells. Importantly, the percentages of live HEK-293T cells treated with hydrogels and the photoirradiation process remained above 90% after 24 h incubation (Fig. 8b, c). In addition, the cells on hydrogels treated with green light (soft hydrogel) adopt a more spread-out and connected morphology compared to the round shaped cell clusters on the non-irradiated hydrogel (stiff hydrogel). This spread-out effect on cell morphology due to softening of the hydrogel substrate was also observed in our photodegradable hydrogels systems[54]. Treatment with hydrogels did not result in significant differences in the percentage of apoptotic cells as compared to the untreated control (Fig. 8c). On the other hand, the cell viability obtained in cells exposed to hydrogels was higher than that of the cells exposed to DMSO.

We further assessed the cytotoxicity of hydrogels using human peripheral blood mononuclear cells (PBMCs) from healthy donors. The majority of PBMCs treated with hydrogels before and after irradiation were alive, achieving a viability of more than 80% (Fig. 8d, e).

As a positive control, treatment with DMSO led to low cell viability, evidenced by the high percentage (>90%) of late apoptotic and necrotic cells (Fig. 8e). Of note, hydrogels before and after irradiation with an LED green light showed similar cell viability in both HEK-293T cells and PBMCs (Fig. 8c and e). In summary, these findings indicate that thioindigo-containing hydrogels – and the photoswitching process – both have good biocompatibility with cells.

## Discussion

Thioindios were discovered more than ten decades ago[35], and their negative photochromism was thoroughly examined in the 1970-1980s[30,37,42,44]. Despite their advantageous switching properties in the visible wavelength region, applications of thioindigo in (bio)materials to date have been hindered by their strong tendency to aggregate in all organic solvents and water. We address this critical shortcoming by introducing a thioindigo bismethacylate that can: (i) be readily synthesised on a gram scale, and (ii) participate in free radical thiol-ene addition for covalent insertion into the polymer backbone via step-growth polymerization. Our detailed investigation of the photo-isomerization – by mapping the isomers' conversion as a function of activation wavelength in a photochemical action plot – reveals the optimal wavelength range for maximum enrichment of the *cis-* or *trans*-isomers at 540-550 nm and 440-470 nm, respectively. It should be noted that our previous work on action plot investigation has mostly focused on photocycloadditions, photothermal hybrid reactions, and photorelease (Norrish type) reactions, all of which display a

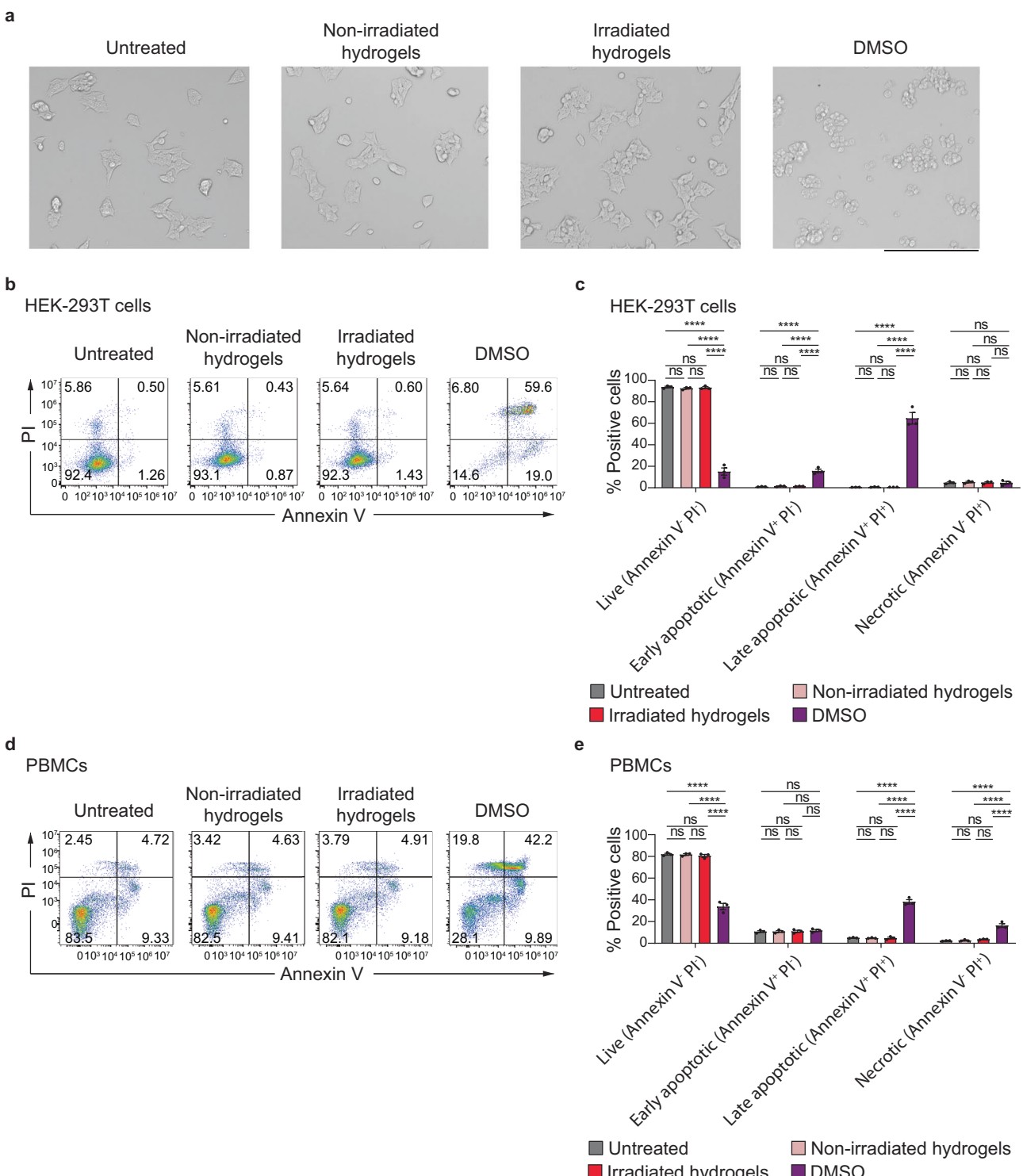

**Fig. 8 | Effect of thioindigo-containing hydrogels and the photoswitching on cell cytotoxicity. a** Representative microscopic images of HKE239T cells treated with **Gel**$_{T5}$ before or after irradiation with an LED green light of $\lambda_{max} = 540$ nm, or with DMSO. Scale bar, 275 μm; **b** Representative flow cytometry plots showing Annexin/PI staining of HEK-293T cells after a 24 hour-treatment with non-irradiated or irradiated hydrogels, or with DMSO; **c** Proportion of viable (Annexin⁻PI⁻), early apoptotic (Annexin⁺PI⁻), late apoptotic (Annexin⁺PI⁺) and necrotic (Annexin⁻PI⁺) HEK-293T cells after 24 h treatment with non-irradiated or irradiated hydrogels, or

with DMSO ($n = 3$ biological independent samples); **d** Representative flow cytometry plots showing Annexin/PI staining in PBMCs after a 24h-treatment with non-irradiated or irradiated hydrogels, or with DMSO; **e** Proportion of viable, early apoptotic, late apoptotic and necrotic PBMCs after 24 h treatment with hydrogels, or with DMSO ($n = 3$ biological independent samples). All bar graphs represent mean with error bars show SEM. ns – not significant, and ****$P < 0.0001$ determined by one-way ANOVA test with Tukey post-hoc test (**c, e**), the exact $p$ values are provided in Supplementary Tables.

red-shift in the photoreactivity compared to their absorption wavelengths[49]. Similarly, we observed a red-shift of the activation wavelength for *trans*- to *cis*- isomerization of thioindigo, and an interesting blueshift for *cis*- to *trans*-conversion wavelength. These findings provide highly reliable guidance to the irradiation window required for the most efficient conversion, compared to the ineffective conventional reliance on UV-Vis absorption spectra of the photoactive isomers.

When incorporated into a PEG polymer mainchain, the thioindio units display reversible photoisomerization by green and blue light, similar to that of the monomer compound. This photoswitching behaviour was observed in a range of organic solvents. Interestingly, the self-aggregation of thioindigo moieties induces physical interaction, and the corresponding gelation of thioindigo-containing PEG, in water at a physiological temperature (37 °C). The reversible photoisomerization further enables modulation of the hydrogel's modulus by alternating green and blue light irradiation. Critically, we demonstrated the insertion of the thioindigo units within covalently crosslinker PEG hydrogel, consequently altering the hydrogels optical and physical properties. The hydrogels' stiffness can be decreased or increased by green or blue light irradiation, respectively. Our preliminary cell testing on the hydrogels indicates that the thioindigo-containing PEG hydrogels and the photoswitching processes are non-harmful to living cells, and thus our hydrogel scaffolds are suitable for further cell-based study. Altogether, these findings demonstrate the potential of thioindigo photoswitch in polymeric materials, offering well-defined spatial and temporal control over the physical properties. Specifically relating to biological research, our all-visible-light responsive hydrogel platform offers significant potential for personalized disease models, opening pathways to gain insights into the effects of stiffness changes on disease progression.

## Methods

### Synthetic procedures
Detailed synthetic procedures are described in the Supplementary Information and are accompanied by reaction schemes and NMR characterizations figures.

### Photoisomerization Action Plot Measurements
The on-line tracking of absorbance spectra during irradiation was conducted using an in-situ UV-Vis apparatus. An Ocean Optics DH-MINI Deuterium-Tungsten-Halogen lamp was coupled via optic fibres (P400-025-SR) to an Ocean Optics FLAME-T-UV-VIS spectrometer, sensitive from 200 to 850 nm, via a cuvette holder. The custom designed cuvette holder enabled laser irradiation into the bottom of the cuvette, while simultaneously measuring absorbance through the side of the cuvette. An Opolette 355 tuneable OPO, emitting 5 ns pulses from 210-2400 nm at a repetition rate of 20 Hz, was directed upwards into the bottom of a quartz fluorescence (Hellma Analytics quartz high precision cell). The laser energy deposited into the sample was measured above the cuvette holder before and after experiments using a Coherent EnergyMax thermopile sensor (J-25MB-LE) to account for any power fluctuations during irradiation.

Thioindigo *trans*-**1** samples were prepared in chloroform with a concentration of 48 μmol·L$^{-1}$ in CHCL$_3$ at 25 °C. Kinetic absorbance data at 540 nm was recorded every 250 ms (50 ms integration time, 5 scan average). In addition, full spectra were saved every 10 s during irradiation. All data processing was performed in Matlab®. The slow switching rates at long wavelengths, combined with the competing thermal relaxation, meant that a constant photon flux was not employed for all wavelengths. Instead, the switching rates (measured in s$^{-1}$) were determined using an exponential fit to kinetic absorption measurements and were normalised by the incident photon flux employed at each wavelength (measured in photons s$^{-1}$) to give a normalised switching rate with units of photon$^{-1}$. For full details refer to the Supplementary Information.

### Photoisomerization action plot measurements
A solution of *trans*-1 in CHCl$_3$ (0.6 mL, $c$ = 48 μmol·L$^{-1}$) was placed in an optically flat glass vial and the vessel was crimp sealed. The irradiation was initiated from below using an Opotek Opolette 355 OPO laser light source, producing 7 ns, 20 Hz pulses with a flattop spatial profile. The output beam was expanded to 6 mm diameter to ensure uniform irradiation across the whole sample volume. The laser beam is positioned to pass through an electronic shutter and directed upwards using a UV silica right angle prism. The laser energy deposited into the sample was measured using a Coherent EnergyMax thermopile sensor (J-25MB-LE) and adjusted to deliver the same number of photons to the reaction mixture at discreet wavelengths.

### Hydrogels fabrication
In a typical procedure, PEG$_{12}$-(SH)$_2$ (7.8 mg, 0.01 mmol, 1 eq. of thiol group) and thioindigo bismethacrylate (0.1-0.5 eq. of methacrylate group, refer to Fig. 5b for the various ratios of thioindigo bismethacrylate used), were dissolved in chloroform (5 mL). AIBN (1 mg, catalytic amount) was added, and the solution was bubbled with nitrogen for 15 min, before being heated at 65 °C under refluxing conditions for 24 h. Chloroform was evaporated under a gentle flow of nitrogen until a viscous oil was obtained. The residue was dissolved in DMF (0.5 mL), and PEG$_{448}$-(propiolate)$_4$ (100 mg, 5·10$^{-3}$ mmol, 1 eq. of propiolate group) was added. The solution was subsequently casted onto a poly(dimethylsiloxane) (PDMS) mould to form square shaped organogel within 3 h of curing at 60 °C. The resultant gel was placed into DI water (500 mL) and allowed to swell over 24 h, with frequent changes of water, to obtain thioindigo-containing hydrogel materials.

### Computational methods
Quantum calculations were performed utilizing Gaussian 16 A. Avogadro software was used for the visualization of molecules and their molecular orbitals. All calculations were performed at the M06-2X/def2-SVP. level of theory with the SMD solvation model in a non-polar environment (dichloromethane), unless mentioned otherwise. We used this nonpolar solvent to mimic the environment in the polymer matrix. The corrected linear solvation (cLR) formalism was used to obtain the electronic energy of various states unless stated otherwise. All the structure was optimized and validated by checking positive vibrational frequencies (except for those optimized during potential energy surface scanning).

During the contraction of the potential energy surface (PES) in both the ground and excited states, we constrained the four dihedral angles around the central bonds, systematically varying them from 0 to 180° at intervals of 10°. All other parameters were optimized freely. For the calculations in the ground state potential energy surface (S$_0$ PES), we utilized the M06-2X/def2SVP level of theory. In the case of the excited state potential energy surface (S$_1$ PES), we conducted geometry optimizations using B3LYP/def2SVP and then performed single-point calculations employing the M06-2X/def2SVP level of theory based on the cLR formalism.

### Rheological studies
Rheological experiments were measured using an Anton Paar Physica rheometer (MCR 302e) with a plate-plate configuration. The lower plate is made of quartz and the upper plate is made of stainless steel with a diameter of 12 mm. A liquid lightguide was placed underneath the lower plate, with the other end connected to a Mightex WheeLED™ wavelength-switchable LED source. In a typical experiment, hydrogel sample (*ca*. 50 mg) was placed on the quartz plate and the upper plate was brought down to a measurement gap of 0.2 mm. A layer of paraffin

oil was applied on the edge of the upper plate to prevent dehydration. The measurement was started by applying a 1% strain with a frequency of 0.1 Hz on the sample.

## Cell cytotoxicity testing

To evaluate if hydrogels and the photoswitching under green light irradiation have potential cytotoxic effects on living cells, we used the human embryonic kidney (HEK-293T) cell line obtained from the American Type Culture Collection (ATCC, USA). We also used human peripheral blood mononuclear cells (PBMCs) isolated by gradient density technique (Ficoll-Paque plus, Cytiva, USA) from whole blood of individuals with informed consent from the Health Sciences Authority of Singapore. HEK293T cells and PBMCs were cultured in DMEM or RPMI 1640 (ThermoFisher Scientific, USA) respectively, supplemented with 10% heat-inactivated FBS (Biosera, USA), and 1% penicillin/streptomycin (ThermoFisher Scientific) and maintained in a humidified incubator at 37 °C with 5% $CO_2$.

Apoptosis of cells was detected using Annexin V-APC (Biolegend, USA) and Propodium Iodide (PI, ThermoFisher Scientific) staining according to the manufacturer's instructions. Briefly, HEK293T cells ($5 \cdot 10^4$ cells/well) or PBMCs ($10^5$ cells per well) were pre-seeded in a 24-well plate for 8 h. Subsequently, a piece of hydrogel of about 4.8 mm$^2$ hydrogels was placed over the culture in each well. Hydrogels were either non-irradiated or irradiated with an LED green light ($\lambda_{max}$ = 540 nm). A negative control (only DMEM or RPMI complete media) and a positive control (complete media containing 20% DMSO) were included to validate the protocol. After 24 h incubation, hydrogels were removed, and microscopic images of cells were taken using an Invitrogen EVOS™ M7000 Imaging System. Subsequently, cells were harvested and washed in cold Annexin V binding buffer, followed by incubation with 100 μL Annexin V binding buffer (BD Pharmingen, USA) containing 5 μL of APC Annexin V and 1 μL of 100 μg/mL PI working solution for 15 min at room temperature in the dark. Then cells were washed with Annexin V binding buffer, and samples were acquired by flow cytometry using the CytoFLEX LX (Beckman Coulter, USA). Thereafter, the data were analysed using the FlowJo software V10 (Three Star, USA) and GraphPad Prism 8 (San Diego,CA, USA).

## Reporting summary

Further information on research design is available in the Nature Portfolio Reporting Summary linked to this article.

# Data availability

The authors declare that the data supporting the findings of this study are available within the article and its Supplementary Information files. Data are available from the corresponding authors upon request.

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

## Acknowledgements

C.B.-K. acknowledges funding by the Australian Research Council (ARC) in the context of a Laureate Fellowship underpinning his photochemical research program as well as QUT's Centre for Materials Science for continued support. P.H.D.N and M. T. N. L. acknowledge financial support of the Singapore Ministry of Health's National Medical Research Council under its Open Fund – Individual Research Grant (NMRC/OFIRG/MOH-000643). H.L. and X.L. acknowledge financial support of Ministry of Education, Singapore (MOE MOET2EP10120-0007) and the Singapore University of Technology and Design (SKI 2021_03_10). This research project was additionally supported by the National Research Foundation (NRF) Singapore under its NRF Investigatorship Scheme (NRF-NRFI07–2021–0003). The authors thank Miss He Xinyang for assistance in preparing the graphics and are grateful for the computing service of SUTD-MIT IDC and the National Supercomputing Centre (Singapore).

## Author contributions

S.L.W. undertook the wavelength-dependent photoisomerization experiments. V.X.T. contributed to the synthesis of small molecules, preparation of the polymers, fabrication and characterisation of hydrogels. H.L. and X.L conducted quantum chemical calculations. P.H.D.N. and M.T.N.L. investigated the biocompatibility of hydrogels and photoswitching processes. X.J.L., C.B.-K. and V.X.T. conceptualised the study and acquired funding. All authors contributed to manuscript preparation and editing.

## Competing interests

The authors declare no competing interests.
