## [Peer Review File · Nature Communications]

Visible Light-Enabled Switching of Soft Material Properties Based on Thioindigo PhotoswitchesREVIEWER COMMENTS

Reviewer #1 (Remarks to the Author):

The manuscript by Loh, Barner-Kowollik, Truong and co-workers reports synthesis and properties of a polymer comprising thioindigo moieties in main chains. Especially important is the achievement of the solubility in water. The development of new photoswitchable materials is a vibrant topic, which perfectly fits to the broad readership of Nat. Commun. Nevertheless, I cannot recommend the manuscript for publication for the following reasons:

- 1) The authors claim: "...we pioneer a strategy towards the integration of the thioindigo function into polymer main chains, enabling the assessment of the photoswitching of the thioindigo-containing macromolecules...". Actually, there is a publication by Irie et al. (<https://iopscience.iop.org/article/10.1143/JJAP.39.L633/meta>) describing the incorporation of thioindigo into the main polymer chains with follow up photoswitching. This study should be cited and compared with the results obtained in the manuscript.
- 2) The photochemical studies are mainly qualitative and do not correspond to the current characterization standards for photoswitchable molecules. Quantum yields for the forward and backward isomerizations should be provided for all applied conditions (monomer, polymer in different solvents).
- 3) The rates of photoisomerization and thermal recovery are significantly dependent on the experimental conditions and are, in general, not very helpful. For thermal backwards isomerization, Gibbs free activation energies should be obtained, which characterize the thermal barrier of the backward reactions. These values can be calculated through the Eyring plot. With the Gibbs free activation energies in hand, one can easily estimate the thermal half-life of the photoinduced species at any temperature. For the publication, in addition to the Gibbs energies, it would be good to provide the values for the thermal half-lives at 25 C.
- 4) Although I completely understand the difficulties regarding the conventional NMR detection of short-living photoinduced species, nowadays this is not a problem anymore. There is a large amount of developed approaches for real-time NMR spectroscopy with in situ irradiation, which allow the detection even of short-living photoreaction intermediates. Please, check the following reviews and references therein: <https://doi.org/10.1002/cptc.201900109>, <https://doi.org/10.1002/cctc.202201583>. Some of these approaches, for example, the one described by Gschwind et al. (<https://doi.org/10.1016/j.jmr.2013.04.011>), are fast, easy and cheap to implement. Alternatively, mathematical methods such as the Fisher method (recently used for indigo derivatives <https://doi.org/10.1002/cptc.201900032>) can be applied to calculate the PSS compositions and the spectra of the photoinduced isomers from the absorption spectroscopy data.
- 5) Figure 3: I am not sure that the spectra obtained in methanol and water indicate photoswitching because their shapes seem not to change at all. Can some different photochemistry in polymer happen, which simply changes the solubility upon irradiation? In any case, it would be helpful to perform DFT calculations of the trans- and cis-forms in corresponding solvents to compare the experimental data with the calculated values of the absorption maxima for both forms.

6) Although the biocompatibility studies are clear, I did not get, for which purpose such a photoswitchable hydrogel can be applied in medicine. Could the authors add some explanations to the text?

Reviewer #2 (Remarks to the Author):

In their manuscript titled "Visible light-enabled switching of soft material properties based on thioindigo photoswitches", Walden et al. react methacrylate-tethered thioindigos with PEG-SH chains and later form crosslinked systems out of these linear polymers, ending up with networks that can be gelled with water. They demonstrate that PEG-thioindigo linear polymers can be photoisomerised in a range of solvents and, when crosslinked, as part of a hydrogel material. In the latter case, the mechanical properties of the hydrogel can be tuned by isomerising the switch from trans to cis or vice versa with green and blue light, respectively.

The authors have done a commendable work in synthesising the thioindigo polymer and in the following material fabrication. The synthesis seems robust and straightforward, and the characterisation of the material has been extended to biocompatibility, which is something most manuscripts do not address in such detail. The manuscript is quite easy to read, and the figures are aesthetically pleasing. There is substantial novelty as well, seeing that this is the first time thioindigos (or to our knowledge any indigoids, for that matter) have been used in this context. However, we feel that the authors have not demonstrated a big enough progress in the field to warrant publication in Nature Communications: certain limitations seem to exist, and comparison to already existing similar systems is lacking. Thus, we believe this work would be better suited for a more specialised journal focusing on functional materials.

Independent of where the work is published, we feel that it could be greatly improved with the following additions and modifications:

- In both the introduction and the discussion/conclusions part, comparison to other photoswitches and photoswitchable hydrogel systems is inadequate. The authors correctly identify the rigidity of thioindigos as a potential advantage over azobenzenes but do not mention other indigoid photoswitch classes that share the same property while having some extra advantages of their own. For instance, hemithioindigos are thermally stable, whereas N-functionalised indigos can be switched with red light, even in a solid environment (see the references below). It would be important to give proper context for thioindigos and why they might be better suitable for this purpose than the other aforementioned indigoids. Similarly, what about already existing hydrogel systems based on azobenzene (there are many examples): how do your results compare to them? Did you get more or less tuneability in the mechanical properties than the state-of-the-art azobenzene systems?

<https://chemistry-europe.onlinelibrary.wiley.com/doi/10.1002/chem.201700826>

<https://pubs.rsc.org/en/content/articlelanding/2023/sc/d2sc06790k>

- The system seems to be really slow, mechanical properties changing over the course of up to an hour. Do the authors have any hypothesis for why this is so slow? Could some ideas be given for how to overcome this problem? Again, how does this compare to existing photoresponsive hydrogel materials?

- The photoisomerisation spectra in Figure 3 only show a clear isosbestic point in chloroform and DMSO. Thus, it seems that in the other solvents there is either degradation or perhaps photoinduced aggregation; something that removes photochromic compounds from the solution. Will you recover the original spectra in time (which would rule out degradation)? Either way, this phenomenon should be discussed and perhaps investigated to determine that the switching is actually reversible.

- No spectral analysis has been carried out for the hydrogels. This would, however, be important in order to know to which extent the thioindigo switches inside the hydrogel. It was recently observed that for N-functionalised indigos, the indigo concentration changes the switching dynamics quite drastically, and it would thus be interesting to know whether a similar phenomenon is observed in your case - or if not, this would be a benefit for thioindigo compared to indigo. It should be rather straightforward to make thin enough hydrogels that could be studied with UV-vis spectroscopy. Then you could see a) what the PSS is like and b) how fast the thermal back-isomerisation is, and whether this happens on the same timescale as the changes in the mechanical properties or not.

- It seems that the mechanical properties do not return to the original when switched for the first time. This is understandable for the two-way photochemical switching (Figure 4a) due to both wavelengths forming PSSs that differ from 100% E or Z, but for the thermal back-isomerisation (Figure 4b) one would expect the mechanical properties to return to the original for a truly reversible system. It seems that some irreversible change occurs on the first switching cycle, after which switching is reversible. This should be discussed and perhaps some rationale given, if possible.

- In the introduction, you mention that the lack of conformational rigidity in azobenzenes arises from the rotational isomerisation pathway (as compared to in-plane inversion). However, it does not matter what the mechanism between the planar trans isomer and the twisted cis isomer is, as the conformational change is dictated by the ground-state minimum of the cis isomer, which is independent of the pathway via which it is reached. Bear in mind that also (thio)indigos isomerise via rotation and still have rigid, planar structures for both isomers.

- A small note: in text, you mention that no observable differences are seen in Figure 2c during multiple isomerisation cycles. There is, however, a clearly (although slightly) decreasing trend in the absorbance values, so I would rephrase along the lines of "no major degradation" or "only slight degradation".

Reviewer #3 (Remarks to the Author):

The authors report visible light-induced switching of soft matter materials properties that the integration of the thioindigo function into polymer main chains. The photoswitching of thioindigo within crosslinked

structures enables visible light induced modulation of the hydrogel stiffness. This material shows excellent cytocompatibility. Their discussion is sound overall, but the authors need to address the questions listed below before the publication is fully warranted.

Comments:

1. The molecule (1) shows excellent reversible trans-/cis-photoswitching (>10 cycles), but why does it show only three changes in the material? Can more times?
2. According to previous works (Nature nanotechnology 10.2 (2015): 161-165; Nature nanotechnology 12.6 (2017): 540-545), light can drive motor to produce a continuous unidirectional out-of-equilibrium rotation, resulting in a macroscopic contraction of the entire network. The mechanism (Fig. 1) proposed by the authors does not seem to explain the problem better. According to the mechanism proposed by the authors, this change in shape should be completely reversible, but it does not seem to be (Rheological data). And there is no other data to support this result, such as pictures.
3. The authors note that the hydrogels stiffness can be modulated by green or blue light irradiation, but only cytotoxicity has been assessed. However, the effect of varying stiffness on cells is unknown. I suggest the author should evaluate this issue, otherwise it is unnecessary.
4. Some obvious mistakes should be avoided.

Such as:

For the reference, first character uppercase in the title should be uniformed, ref 1-4, 11, 14, 16, 19-20, 23-26, 28-31, 33-36, 41, 43.

Format of the symbols, including Mn, Tg, and λ_{\max} (italic).

3. Some related works about visible light-induced photoswitchable materials are recommended to be referenced. For example, Adv. Funct. Mater., 2023, 2303765. Angew. Chem. Int. Ed. 2020, 59, 18532 and J. Am. Chem. Soc. 2020, 142, 7995.

Reviewer #1 (Remarks to the Author):

The manuscript by Loh, Barner-Kowollik, Truong and co-workers reports synthesis and properties of a polymer comprising thioindigo moieties in main chains. Especially important is the achievement of the solubility in water. The development of new photoswitchable materials is a vibrant topic, which perfectly fits to the broad readership of Nat. Commun. Nevertheless, I cannot recommend the manuscript for publication for the following reasons:

Response: We thank the reviewer for the kind assessment of our report, and the remark on the importance of new photoswitchable macromolecules that are water-soluble. We have included extensive additional experiments to address the concerns raised in the detailed comments and highlight the significance of our work on the applications of thioindigo phototoswitch in soft matter materials.

1) The authors claim: "...we pioneer a strategy towards the integration of the thioindigo function into polymer main chains, enabling the assessment of the photoswitching of the thioindigo-containing macromolecules...". Actually, there is a publication by Irie et al. (<https://iopscience.iop.org/article/10.1143/JJAP.39.L633/meta>) describing the incorporation of thioindigo into the main polymer chains with follow up photoswitching. This study should be cited and compared with the results obtained in the manuscript.

Response: We thank the reviewer for referring to a related report of thioindigo photoswitching in polymer thin film that was overlooked. We have included this reference in the introduction (page 3) and discussion (page 6) of our report.

2) The photochemical studies are mainly qualitative and do not correspond to the current characterization standards for photoswitchable molecules. Quantum yields for the forward and backward isomerizations should be provided for all applied conditions (monomer, polymer in different solvents).

Response: As per the reviewer's suggestion, we have enhanced the photochemical studies to better align with current characterisation standards. In particular, we have replaced the switching rate in Figure 2d with the quantum yield for forward and backward isomerisation. We have also provided individual experimental data of the photoswitching at discrete wavelength in the 370-530 nm wavelength range

into the supporting information (Section 1.5) as well as a detailed description of how the quantum yields were determined.

3) The rates of photoisomerization and thermal recovery are significantly dependent on the experimental conditions and are, in general, not very helpful. For thermal backwards isomerization, Gibbs free activation energies should be obtained, which characterize the thermal barrier of the backward reactions. These values can be calculated through the Eyring plot. With the Gibbs free activation energies in hand, one can easily estimate the thermal half-life of the photoinduced species at any temperature. For the publication, in addition to the Gibbs energies, it would be good to provide the values for the thermal half-lives at 25 C.

Response: We thank the reviewer for their comment. To obtain a universal data set on the photoisomerization process, we have performed quantum chemical calculations (results are provided on page 7-10 and Fig. 5), obtaining an energy barrier of 0.423 eV for the *trans*- to *cis*- transformation and an energy barrier of 0.37 eV for the *cis*- to *trans*-conversion. Our calculations further account for the hydrogen bonding energy in protic solvents (such as methanol and water), explaining the lower conversion efficiency in such solvents. As requested, we have also reported the experimentally determined thermal half-life at 24 °C in the revised manuscript.

4) Although I completely understand the difficulties regarding the conventional NMR detection of short-living photoinduced species, nowadays this is not a problem anymore. There is a large amount of developed approaches for real-time NMR spectroscopy with in situ irradiation, which allow the detection even of short-living photoreaction intermediates. Please, check the following reviews and references therein: <https://doi.org/10.1002/cptc.201900109>, <https://doi.org/10.1002/cctc.202201583>. Some of these approaches, for example, the one described by Gschwind et al. (<https://doi.org/10.1016/j.jmr.2013.04.011>), are fast, easy and cheap to implement. Alternatively, mathematical methods such as the Fisher method (recently used for indigo derivatives <https://doi.org/10.1002/cptc.201900032>) can be applied to calculate the PSS compositions and the spectra of the photoinduced isomers from the absorption spectroscopy data.

Response: We completely agree with the reviewer that an in situ-NMR is the optimal way to characterise these short-lived species. In fact, we are currently in the process of commissioning a laser-coupled NMR at QUT that is expected to be operational later this year. We disagree, however, that such a system, when executed thoroughly, is neither fast, cheap nor easy to install. The purpose of the NMR

measurements in the current work was not to determine the photostationary state, but rather to confirm the structure of the isomers and that no degradation or side products were being formed during irradiation. We acknowledge that our reporting of these results was not ideal and have clarified the relevant text in the manuscript.

5) Figure 3: I am not sure that the spectra obtained in methanol and water indicate photoswitching because their shapes seem not to change at all. Can some different photochemistry in polymer happen, which simply changes the solubility upon irradiation? In any case, it would be helpful to perform DFT calculations of the *trans*- and *cis*-forms in corresponding solvents to compare the experimental data with the calculated values of the absorption maxima for both forms.

Response: Based on the suggestion of the reviewer, we have conducted quantum chemical calculations of thioindigo photoswitching. These calculations indicate that the hydrogen bonding energy in protic solvents significantly affects the relative Gibbs free energy of both the *trans*- and *cis*-isomers, consequently altering their λ_{abs} . Based on our calculations, we also find that the *cis*-form is more polar than the *trans*-form. To confirm that photoswitching does occur, we have performed up to 10 cycles of alternating irradiation of green and blue light on the polymer solutions in methanol and water. Our results (Fig. S14) indicate repeatable photoswitching between the *cis*- and *trans*-states and no significant photodegradation occurring during light irradiations.

6) Although the biocompatibility studies are clear, I did not get, for which purpose such a photoswitchable hydrogel can be applied in medicine. Could the authors add some explanations to the text?

Response: As demonstrated in our study, the hydrogels are biocompatible and can be used as a scaffold for the study of dynamic cell-materials interaction i.e., mechanotransduction of cells and modelling of disease progression. Indeed, we observed different cell morphologies when the hydrogel substrate was softened by green light irradiation, compared to cells grown on stiff hydrogels with no green light treatment. To reduce ambiguity, we have removed the 'medicinal' context and included additional texts in the discussion to explain the potential of our photoswitchable hydrogels in cells materials interaction and disease modelling.

Reviewer #2 (Remarks to the Author):

In their manuscript titled "Visible light-enabled switching of soft material properties based on thioindigo photoswitches", Walden et al. react methacrylate-tethered thioindigos with PEG-SH chains and later form crosslinked systems out of these linear polymers, ending up with networks that can be gelled with water. They demonstrate that PEG-thioindigo linear polymers can be photoisomerised in a range of solvents and, when crosslinked, as part of a hydrogel material. In the latter case, the mechanical properties of the hydrogel can be tuned by isomerising the switch from trans to cis or vice versa with green and blue light, respectively.

The authors have done a commendable work in synthesising the thioindigo polymer and in the following material fabrication. The synthesis seems robust and straightforward, and the characterisation of the material has been extended to biocompatibility, which is something most manuscripts do not address in such detail. The manuscript is quite easy to read, and the figures are aesthetically pleasing. There is substantial novelty as well, seeing that this is the first time thioindigos (or to our knowledge any indigoids, for that matter) have been used in this context. However, we feel that the authors have not demonstrated a big enough progress in the field to warrant publication in *Nature Communications*: certain limitations seem to exist, and comparison to already existing similar systems is lacking. Thus, we believe this work would be better suited for a more specialised journal focusing on functional materials. Independent of where the work is published, we feel that it could be greatly improved with the following additions and modifications.

Response: We thank the reviewer for the kind assessment regarding the novelty of our research, and the suggestions to improve our report. We hope the additional experiments and quantum chemical communications offer considerable insight into the systems, and new progresses that are suitable for publication in *Nature Communications*.

Comment: In both the introduction and the discussion/conclusions part, comparison to other photoswitches and photoswitchable hydrogel systems is inadequate. The authors correctly identify the rigidity of thioindigos as a potential advantage over azobenzenes but do not mention other indigoid photoswitch classes that share the same property while having some extra advantages of their own. For instance, hemithioindigos are thermally stable, whereas N-functionalised indigos can be switched with red light, even in a solid environment (see the references below). It would be important to give proper context for thioindigos and why they might be better suitable for this purpose than the other

aforementioned indigoids. Similarly, what about already existing hydrogel systems based on azobenzene (there are many examples): how do your results compare to them? Did you get more or less tuneability in the mechanical properties than the state-of-the-art azobenzene systems?

<https://chemistry-europe.onlinelibrary.wiley.com/doi/10.1002/chem.201700826>

<https://pubs.rsc.org/en/content/articlelanding/2023/sc/d2sc06790k>

Response: As suggested, we have included the references on indigos and hemithioindigos in the introduction with relevant texts, as well as the comparison of our results with azobenzene-containing hydrogels in the Results section.

- The system seems to be really slow, mechanical properties changing over the course of up to an hour. Do the authors have any hypothesis for why this is so slow? Could some ideas be given for how to overcome this problem? Again, how does this compare to existing photoresponsive hydrogel materials?

Response: The time required for the tuning of mechanical properties relies on several factors. The first is the isomerisation rate, which we have already shown in Figure 2 to be reasonably efficient. The second is the penetration of light throughout the material volume. The molar absorptivity of both the *trans*- and *cis*-isomers is high (close to $10^4 \text{ L mol}^{-1} \text{ cm}^{-1}$) and so is the thioindigo incorporation into the hydrogel (close to 5.5 wt%). These two factors combine to significantly limit the light penetration (and hence mechanical tuning) throughout the material. We have made efforts to minimise the impact of these effects by irradiating with wavelengths offset from the absorption maxima. Further increases in the mechanical tuning rates are expected to be possible by selecting irradiation sources even further removed from the absorption maxima. In addition, as indicated by our chemical quantum calculations, the photoswitching is highly constrained in protic solvents – especially water – which explains the slow photoswitching kinetics of the polymer network. We have included the explanation and relevant comparison with azobenzene-containing hydrogels (ref. 46, *Biomacromolecules* **2015**, *16*, 798) in our manuscript.

- The photoisomerisation spectra in Figure 3 only show a clear isosbestic point in chloroform and DMSO. Thus, it seems that in the other solvents there is either degradation or perhaps photoinduced aggregation; something that removes photochromic compounds from the solution. Will you recover the original spectra in time (which would rule out degradation)? Either way, this phenomenon should be discussed and perhaps investigated to determine that the switching is actually reversible.

Response: We thank the reviewer for the feedback on the photoisomerization spectra. We agree the decrease in the 540 nm absorbance band is due to photoinduced aggregation of the polymer chains. After green/blue light treatment the spectra did not revert to original spectra in all polar solvents, even with prolonged blue light exposure. To rule out any photodegradation effects, we carried out green/blue light irradiations for up to 10 cycles on the polymer solutions in MeOH and water. The results (Fig. S14) show that photoswitching can be induced by green/blue light and no significant photodegradation occurring during the light irradiation processes.

- No spectral analysis has been carried out for the hydrogels. This would, however, be important in order to know to which extent the thioindigo switches inside the hydrogel. It was recently observed that for N-functionalised indigos, the indigo concentration changes the switching dynamics quite drastically, and it would thus be interesting to know whether a similar phenomenon is observed in your case - or if not, this would be a benefit for thioindigo compared to indigo. It should be rather straightforward to make thin enough hydrogels that could be studied with UV-vis spectroscopy. Then you could see a) what the PSS is like and b) how fast the thermal back-isomerisation is, and whether this happens on the same timescale as the changes in the mechanical properties or not.

Response: We thank the reviewer for the feedback. We have thus undertaken the measurement of the UV-Vis absorbance of hydrogel **GeIT5** in a cuvette as suggested and observed the photoswitching similar to the changes observed in solution. We have also included the comparison with other solid polymer systems that contain thioindigo or indigo (ref 40: *Japanese Journal of Applied Physics* **2000**, *39* (6B), L633).

- It seems that the mechanical properties do not return to the original when switched for the first time. This is understandable for the two-way photochemical switching (Figure 4a) due to both wavelengths forming PSSs that differ from 100% E or Z, but for the thermal back-isomerisation (Figure 4b) one would expect the mechanical properties to return to the original for a truly reversible system. It seems that some irreversible change occurs on the first switching cycle, after which switching is reversible. This should be discussed and perhaps some rationale given, if possible.

Response: We thank the reviewer for their insight. Indeed, the hydrogel stiffness did not reverse to the original value, possibly due to the stabilizing effect of solvent on the *trans*- isomer. Of note, similar phenomenon was also observed in azobenzene-containing hydrogels (*Biomacromolecules* **2015**, *16*,

798). We have included the noted system in the discussion of the revised manuscript and relevant references relating to the azobenzene hydrogel.

- In the introduction, you mention that the lack of conformational rigidity in azobenzenes arises from the rotational isomerisation pathway (as compared to in-plane inversion). However, it does not matter what the mechanism between the planar trans isomer and the twisted cis isomer is, as the conformational change is dictated by the ground-state minimum of the cis isomer, which is independent of the pathway via which it is reached. Bear in mind that also (thio)indigos isomerise via rotation and still have rigid, planar structures for both isomers.

Response: We thank the reviewer for the helpful remark on the photoswitching mechanism of (thio)indigo and have noted this point in the introduction.

- A small note: in text, you mention that no observable differences are seen in Figure 2c during multiple isomerisation cycles. There is, however, a clearly (although slightly) decreasing trend in the absorbance values, so I would rephrase along the lines of "no major degradation" or "only slight degradation".

Response: We have edited the texts as suggested.

Reviewer #3 (Remarks to the Author):

The authors report visible light-induced switching of soft matter materials properties that the integration of the thioindigo function into polymer main chains. The photoswitching of thioindigo within crosslinked structures enables visible light induced modulation of the hydrogel stiffness. This material shows excellent cytocompatibility. Their discussion is sound overall, but the authors need to address the questions listed below before the publication is fully warranted.

Comments:

1. The molecule (1) shows excellent reversible trans-/cis-photoswitching (>10 cycles), but why does it show only three changes in the material? Can more times?

Response: We have repeated the photo-rheology study to determine the change in G' value for GelT5 up to 10 cycles of green/blue light irradiation and included the data in the SI (Fig. S15).

2. According to previous works (Nature nanotechnology 10.2 (2015): 161-165; Nature nanotechnology 12.6 (2017): 540-545), light can drive motor to produce a continuous unidirectional out-of-equilibrium rotation, resulting in a macroscopic contraction of the entire network. The mechanism (Fig. 1) proposed by the authors does not seem to explain the problem better. According to the mechanism proposed by the authors, this change in shape should be completely reversible, but it does not seem to be (Rheological data). And there is no other data to support this result, such as pictures.

Response: We thank the reviewer for the feedback. The photoswitching of our hydrogels was undertaken in an aqueous environment (highly polar and protic solvent), in contrast to the noted references where the photoswitching of the gels was carried out in acetonitrile (low polar and non-protic solvent). Our results demonstrate that the hydrogels expanded and become more transparent upon green light exposure (refer to Fig. 7a for the changes in hydrogels optical property that can be observed with naked eyes). We have modified Fig. 7a to clearly show the increase in hydrogel's size due to swelling. Our quantum computational calculations indicate the *cis*-isomer ($\mu = 5.87$ D) has a higher dipole moment compared to the *trans*-isomer ($\mu = 4.39$ D), explaining the increase in water sorption (due to the increase in polarity of the *cis*-enriched hydrogel network). We have included these results in the manuscript and include the related references that the reviewer mentioned.

3. The authors note that the hydrogels stiffness can be modulated by green or blue light irradiation, but only cytotoxicity has been assessed. However, the effect of varying stiffness on cells is unknown. I suggest the author should evaluate this issue, otherwise it is unnecessary.

Response: As observed from our results, the cells seeded on the hydrogels after green light treatment displayed a change in the morphology: they became more elongated and spread out. This change may be due to the softening effect of green light irradiation, as shown from the photo-rheology data. The same effect on cells morphology was also observed in our photodegradable hydrogel systems (Adv. Mat., 2021, 202102184.). We have included this observation and comparison in the manuscript.

4. Some obvious mistakes should be avoided. Such as: For the reference, first character uppercase in the title should be uniformed, ref 1-4, 11, 14, 16, 19-20, 23-26, 28-31, 33-36, 41, 43. Format of the symbols, including Mn, Tg, and λ_{\max} (italic).

Response: We thank the reviewers for the detailed feedback, we have corrected the formatted accordingly.

3. Some related works about visible light-induced photoswitchable materials are recommended to be referenced. For example, Adv. Funct. Mater., 2023, 2303765. Angew. Chem. Int. Ed. 2020, 59, 18532 and J. Am. Chem. Soc. 2020, 142, 7995.

Response: We have included the noted references in the introduction as suggested.

REVIEWERS' COMMENTS

Reviewer #1 (Remarks to the Author):

The revised manuscript by Loh, Barner-Kowollik, Truong and co-workers has significantly improved. The authors have considered most concerns carefully and I would like to thank the authors for this. There is only one thing I would like the authors to improve. The new Figure 2 is good but it would be helpful to see the numbers for the quantum yields in the text (e.g. forward reaction, max quantum yield ...% upon irradiation at ... nm, backward reaction, max quantum yield ...% upon irradiation at ... nm). Same applies to the conversion ratio, please mention in the text the max and min trans/cis percentage ranges for the forward and backward photoisomerization.

After this is done, the manuscript can be accepted for publication.

Reviewer #2 (Remarks to the Author):

I would like to thank the authors for substantially improving their manuscript in response to the comments raised by the reviewers. I have no further comments, and believe that the manuscript can be accepted in its present form. Congratulations for the good work!

Reviewer #3 (Remarks to the Author):

In my opinion, the authors have successfully clarified all the points raised in this revision. Then, this paper can be considered for publication as it is.

RESPONSES TO REVIEWERS' COMMENTS

NCOMMS-23-34953

Reviewer #1 (Remarks to the Author):

The revised manuscript by Loh, Barner-Kowollik, Truong and co-workers has significantly improved. The authors have considered most concerns carefully and I would like to thank the authors for this. There is only one thing I would like the authors to improve. The new Figure 2 is good but it would be helpful to see the numbers for the quantum yields in the text (e.g. forward reaction, max quantum yield ...% upon irradiation at ... nm, backward reaction, max quantum yield ...% upon irradiation at ... nm). Same applies to the conversion ratio, please mention in the text the max and min trans/cis percentage ranges for the forward and backward photoisomerization.

After this is done, the manuscript can be accepted for publication.

Response: We thank the reviewer for the kind assessment of our report. We have made the changes as suggested.

Reviewer #2 (Remarks to the Author):

I would like to thank the authors for substantially improving their manuscript in response to the comments raised by the reviewers. I have no further comments, and believe that the manuscript can be accepted in its present form. Congratulations for the good work!

Response: We thank the reviewer for the kind assessment of our report.

Reviewer #3 (Remarks to the Author):

In my opinion, the authors have successfully clarified all the points raised in this revision. Then, this paper can be considered for publication as it is.

Response: We thank the reviewer for the kind assessment of our report.